# Changes in Nurse Staffing Grades of Korean Hospitals during COVID-19 Pandemic

**DOI:** 10.3390/ijerph18115900

**Published:** 2021-05-31

**Authors:** Young-Taek Park, JeongYun Park, Ji Soo Jeon, Young Jae Kim, Kwang Gi Kim

**Affiliations:** 1HIRA Research Institute, Health Insurance Review & Assessment Service (HIRA), Wonju-si 26465, Korea; pyt0601@hira.or.kr; 2Department of Clinical Nursing, University of Ulsan, Seoul 05505, Korea; 3Department of Biomedical Engineering, College of Health Science, Gachon University, Gil Medical Center, Incheon 21565, Korea; jeon1923@naver.com (J.S.J.); youngjae@gachon.ac.kr (Y.J.K.); kimkg@gachon.ac.kr (K.G.K.)

**Keywords:** nurse staffing, nurse staffing level, nurse staffing grades, nurse–patient ratio, nursing shortage

## Abstract

The global COVID-19 pandemic is creating challenges to manage staff ratios in clinical units. Nurse staffing level is an important indicator of the quality of care. This study aimed to identify any changes in the nurse staffing levels in the general wards of hospitals in Korea during the COVID-19 pandemic. The unit of analysis was the hospitals. This longitudinal study observed the quarterly change of the nurse staffing grades in 969 hospitals in 2020. The nurse staffing grades ranged from 1 to 7 according to the nurse–patient ratio measured by the number of patients (or beds) per nurse. The major dependent and independent variables were the change of nurse staffing grades and three quarterly observation points being compared with those during the 1st quarter (1Q) of 2020, respectively. A generalized linear model was used. Unexpectedly, the nurse staffing grades significantly improved (2Q: RR, 27.2%; 95% confidence interval (CI), 15.1–27.6; *p* < 0.001; 3Q: RR, 95% CI, 20.2%; 16.9–21.6; *p* < 0.001; 4Q: RR, 26.6%; 95% CI, 17.8–39.6; *p* < 0.001) quarterly, indicating that the nurse staffing levels increased. In the comparison of grades at 2Q, 3Q, and 4Q with those at 1Q, most figures improved in tertiary, general, and small hospitals (*p* < 0.05), except at 3Q and 4Q of general hospitals. In conclusion, the nurse staffing levels did not decrease, but nursing shortage might occur.

## 1. Introduction

Nurses in hospital wards are responsible for monitoring inpatients for 24 h and providing nursing care without any delay [1]. Their roles cannot be overemphasized. Although many hospitals are already using several advanced technologies and automated devices, the role of hospital nurses has not changed considerably and is, even more, the headline of health care. With the shortage of nurses and the lack of nursing care, patients are at risk of being exposed to dangerous situations [2,3].

The effects of nurse staffing level and appropriate level of nurse allocation on the quality of nursing care in hospital wards have already been extensively reported [4,5,6]. According to a Korean study, patient mortality rates on pneumonia and hemorrhagic septicemia after operations significantly decrease if the nurse staffing level increases [7,8]. Thus, maintaining the appropriate level of nurse staffing and hiring experienced nurses are important [9].

In December 2019, an unknown flu was reported in Wuhan, China; on 20 January 2020, Korea confirmed its first case of COVID-19 [10]. The COVID-19 pandemic has become widespread since its first observation. During this pandemic, the number of patients and workloads of nurses increase; thus, many nurses are needed. Nurses should take care of the patients and monitor their families visiting the hospital at all times. However, many nurses felt heavy workloads and physical and mental burdens during this pandemic [11,12,13]. As a result, many nurses might move out or quit their job, inevitably resulting in a shortage of nurses [14,15,16].

Healthcare facilities such as hospitals and nursing homes in the United States are already experiencing nursing shortages during the COVID-19 pandemic [17,18]. Intensive care units (ICUs) and isolation wards in other countries also experience nursing shortages [19,20]. However, the nurse staffing level in general wards during this pandemic has not yet been investigated using empirical data at the national level in Korea. Thus, monitoring and following up on the changes in the nurse staffing level after the outbreak of the COVID-19 pandemic are necessary.

This study aimed to identify any changes in the nurse staffing levels of hospital wards during the COVID-19 pandemic in Korea. By using the national-level data, this study would provide valid results. The study results would also contribute to the knowledge expansion of international colleagues and provide the opportunity of evaluating the case of Korea regarding the environmental situation of nurse professional groups and any changes or shortages in the nurse staffing level.

## 2. Materials and Methods

### 2.1. Study Design

The unit of analysis is hospitals with over 30 beds. Korea has three hospital types: tertiary hospital, general hospital, and hospitals. Tertiary hospitals have several preconditions, such as being a university hospital or having seven to nine medically specialized medical departments. General hospitals have more than 100 beds. Meanwhile, hospitals have 30 to 100 beds (hereafter, “small hospitals” to differentiate the general term “hospitals”). This longitudinal study observed four timestamps of the four quarters of 2020: 1Q, 2Q, 3Q, and 4Q. The data used in this study were from 1 January 2020, to 31 December 2020. On 19 February 2021, the Institutional Review Board (IRB) of the University of Ulsan approved this study (IRB#: 2021R0016-001) having the study period from the approval date of IRB to 18 February 2022. This study was exempt from formal review by the IRB. 

### 2.2. Data Sources

This study used two data sources. The data were publicly available. For data security concern, the data on nurse staffing grades were directly obtained from the health insurance administrative data provided by the Health Insurance Review and Assessment Service (HIRA), which is a third-party administrative institution that provides professional health insurance claim review and assessment supporting the national health insurance systems of Korea. Furthermore, the COVID-19 outbreak data were from the Korean public data portal of the Ministry of Health and Welfare (MOHW). This portal has provided the national outbreak data of people confirmed with COVID-19, with real-time manners geographically since January 2020 (Figure 1).

### 2.3. Outcome Variables and Independent Variables

This study had three outcome variables, which were the differences of nurse staffing grades between 2Q and 1Q, 3Q and 1Q, 4Q and 1Q. The nurse staffing grades were obtained from the national health insurance administrative dataset, which had been collected by the HIRA. Since November 1999, the MOHW of Korea has introduced the sliding scale of the nursing reimbursement fee for every inpatient care depending on the nurse staffing grades [21]. The grades ranged from 1 to 7 and were calculated by using the nurse-to-patient (or bed) ratio (“NP ratio”) of hospitals. For example, the grades of a hospital with over 100 beds range from grade 1 with an NP ratio of <1: 2.5 to grade 7 with a ratio of ≥1: 6.0 or not reporting their nurse staffing levels to HIRA. In detail, the hospitals obtain grades 1, 2, 3, 4, 5, 6, and 7 if the NP ratios were 1: <2.5, 1: 2.5–3.0, 1: 3.0–3.5, 1: 3.5–4.0, 1: 4.0–4.5, 1: 4.5–6.0, and 1: >6.0 or not reporting, respectively. Thus, grade 1 means high nurse staffing level and grade 7 represents low nurse staffing level. In Korea, the number of beds has been applied for Seoul, the Capital of South Korea, whereas the number of patients has been applied for the rest of the areas since 2018. Generally speaking, the NP ratio or patient-to-nurse ratio is a good indicator of the nurse staffing level [18,22,23]. In Korea, the grades critically affect the inpatient reimbursement fee per patient per day. If some hospitals get the grade 1, then they will get the full amount of inpatient reimbursement fee with some additional monetary incentives. Thus, every hospital has taken care of nurse staffing grades to attain the full amount of inpatient reimbursement fee. Regarding hospitals having the grade 7, this study separately analyzed them because there were groups of hospitals not reporting nurse staffing levels to HIRA. 

This study had several independent variables on various organizational characteristics of hospitals. These variables were the number of hospital beds, years of operation, types of hospital ownership (private vs. nonprofit organization), multihospital systems, location of the facility (urban vs. rural), number of physicians, number of nurses, and the NP ratio difference between 2Q and 1Q, 3Q and 1Q, 4Q and 1Q. These variables were used according to several previous studies [1,7,8].

### 2.4. Statistical Analysis

This study investigated the changes in the nurse staffing level after the outbreak of the COVID-19 pandemic. The outcome variables were measured thrice (2Q–1Q, 3Q–1Q, and 4Q–1Q). Thus, the main outcome variable has a numeric scale. As mentioned, Korea has three hospital types, which have individual organizational characteristics. Thus, this study evaluated the general characteristics according to hospital type. To compare these three groups, we cross-tabulated the data by hospital type and analyzed the basic characteristics of hospitals through the analysis of variance between groups. Before the main analysis, we checked the correlations between the independent variables to identify any multicollinearity issues among the independent variables. To deal with multicollinearity issues, we used separate models. For the main analysis, a generalized linear model was used with 95% confidence intervals for each independent variable. The main data were analyzed using the SAS version 9.1.

## 3. Results

### 3.1. General Characteristics of the Study Subjects

Table 1 lists the general characteristics of the study hospitals. The number of people with COVID-19 quarterly was not the number of hospital patients with COVID-19; instead, it was the number of people with COVID-19 in the areas where each hospital was located. Quarterly nurse staffing levels were the highest in tertiary hospitals and the lowest in small hospitals. In each hospital type, the nurse staffing level after the 1Q increased rather than decreased, indicating that fewer nurses were working at each hospital type after the 1Q. The number of patients with COVID-19 also drastically increased after the 1Q.

Figure 2a presents the quarterly changes in nurse staffing grades and monthly number of patients with COVID-19. Although the number of patients with COVID-19 drastically increased, the quarterly nurse staffing grades slightly decreased, indicating the increase of the nurse staffing level per patient. This result was the basis of the reported results wherein the number of grade 7 hospitals decreased in Figure 2b. Generally, grade 7 includes hospital groups that do not report their nurse staffing levels to HIRA.

### 3.2. Changes in Nurse Staffing Grades during the COVID-19 Pandemic in All Hospitals

Changes in nurse staffing grades in all hospitals after the outbreak of the COVID-19 pandemic, that is, the 1Q of 2020. Depending on observation quarters, nursing staffing grades improved by approximately 20% after the 1Q, indicating an increase in the nursing quality (2Q: RR, 27.2%; 15.1–27.6; *p* < 0.001; 3Q: RR, 20.2%, 16.9–21.6, *p* < 0.001; 4Q: RR, 29.6%; 17.8–39.6, *p* < 0.001) (Table 2).

### 3.3. Changes in Nurse Staffing Grades during the COVID-19 Pandemic in Tertiary Hospitals

Table 3 presents the change in nurse staffing levels in tertiary hospitals after the outbreak of the COVID-19 pandemic (1Q) in 2020. In 3Q and 4Q, the nurse staffing grade significantly improved compared with that in the 1Q of 2020 (3Q: RR, 21.2%; 4.2–35.1, *p* = 0.018; 4Q: RR, 19.3%; 1.9–33.6, *p* = 0.033). Thus, the number of nurses working at tertiary hospitals decreased, possibly because each nurse should take care of fewer patients.

### 3.4. Changes in Nurse Staffing Grades during the COVID-19 Pandemic in General Hospitals

Table 4 shows the change of nurse staffing levels in general hospitals after the 1Q of 2020. The Nurse staffing grade improved by 24.2% at 2Q compared with that at 1Q. Hence, the number of nurses working at general hospitals increased (RR, 24.2%; 2.7–40.9; *p* = 0.030). However, the NP ratio of 3Q and 4Q did not significantly differ from that of 1Q.

### 3.5. Changes in Nurse Staffing Grades during the COVID-19 Pandemic in Small Hospitals

Table 5 summarizes the changes in nurse staffing levels in small hospitals after the 1Q of 2020. The nurse staffing level of hospitals at 2Q–4Q significantly improved after the 1Q. Thus, the nurse staffing level in small hospitals did not decrease (2Q: RR, 29.0%; 13.8–41.5, *p* = 0.001; 3Q: RR, 21.6%; 4.9–35.3, *p* = 0.013; 4Q: RR, 32.9%; 18.6–44.7, *p* < 0.001). For instance, the nurse staffing grade improved by 32.9% at 4Q compared with that at 1Q, implying that each nurse took care of fewer patients.

## 4. Discussion

This study aimed to identify any changes in the nurse staffing levels, especially in the general wards of hospitals, after the outbreak of the COVID-19 pandemic. We adopted a longitudinal study design and used large health insurance administrative data publicly available in Korea. Results showed no suspicious sign of a decrease in nurse staffing levels during the COVID-19 pandemic. Unexpectedly, the hiring status of nurses slightly improved despite the drastic increase in the number of patients with COVID-19.

For all hospitals, the nurse staffing grades improved at 2Q, 3Q, and 4Q compared with that at 1Q, indicating that the number of hospital ward nurses increased. This study result is remarkable. The COVID-19 pandemic would require more nurses in the ICUs [24], especially in large hospitals; thus, some nurses would be reallocated from general wards to ICUs, thereby reducing the nurse staffing level in general wards. However, the study found no significant decrease in the nurse staffing level in general wards. This result may be explained by two points. First, given that hospitals might need to move nurses with higher experiences to the ICUs, they might accept nurses from out-of-hospital boundaries to make up for or complement the nurse shortage in the ICUs. If hospitals need to reduce nurses in general wards and increase nurses in ICUs, they would receive a reduced reimbursement amount regarding the nurse staffing level. Consequently, as the second point, the number of general ward nurses might only be slightly reduced. Thus, hospitals would choose an alternative action recruiting additional nurses for ICUs [25]. However, this kind of explanation would also be a conjecture, thereby requiring further research at the national level. Any increased reimbursement insurance policies on ICUs regarding the COVID-19 pandemic. In addition, some rare possibilities of reduced inpatients could improve the NP ratio.

Moreover, the study results of tertiary hospitals are similar to those of all hospitals without a decrease in nurse staffing levels. Generally, most tertiary hospitals have well-equipped ICUs and experienced nurse staff. The Korean government takes various actions to tertiary hospitals to actively participate in the care for patients with COVID-19 [26]. Thus, the COVID-19 pandemic might critically affect tertiary hospitals in a way similar to the reasons mentioned above. However, the nurse staffing level of tertiary hospitals even increases the nurse staffing level. In Korea, many nurses want to work at tertiary hospitals, which have long waiting lists of nurse applicants [27]. Tertiary hospitals could easily overcome nursing shortages by additionally hiring nurses and reallocating experienced nurses to ICUs [28].

For general and small hospitals, the nurse staffing level improved, except at 3Q and 4Q of general hospitals. Interestingly, the nurse staffing level of small hospitals considerably improved after the outbreak of the COVID-19 pandemic. This result could be logically explained by the high possibility of lack of connection between COVID-19 and small hospitals because small hospitals tended to have no ICUs. Thus, small hospitals might have a further chance of hiring more nurses or low number of patients. Unlike that in small hospitals, the staffing level in general hospitals did not change, except at 2Q, probably because general hospitals would not have many rooms to take care of patients with COVID-19 compared with the tertiary hospitals [29]. Thus, general hospitals would have a mediocre role in standpoint, resulting in no change of staffing levels in general hospitals.

Regarding the hospitals having nurse staffing grade 7 or not reporting their nurse staffing levels to HIRA, this study found that those groups of hospitals also had a tendency of decreasing from Q1 to Q4. Although we considered this fact, this study found that the nurse staffing level did not decrease after the outbreak of the COVID-19 pandemic. Generally speaking, the exact figure of the nurse employment rate of Korea has not been known for, but the number of active nurses was 6.9 per 1000 populations which was lower than the average number of the Organisation for Economic Co-operation and Development (OECD), 9.0, in 2017 [30]. 

This study concluded that despite the severity of the COVID-19 pandemic, the government of Korea took good care of nurse staffing levels in general wards and managed the pandemic well through using various actions and strategies. Effective actions and strategies would be the use of COVID-19 diagnostic test kits, coordination of COVID-19 case assignments to designated hospitals, and mandatory mask-wearing [27]. Another strategy would be introducing telemedicine [31,32], which demonstrated similar efficacy with the face-to-face care tool [33]. According to a report on how professional nurses responded to the COVID-19 pandemic in Korea, the Korean Nurses Association, nurse leaders, and volunteer nurses showed numerous systematic actions against COVID-19 [34]. Additionally, individual hospitals took specific actions against the COVID-19 pandemic by themselves [35,36,37]. Many nurses also showed endless efforts against the pandemic [38]. These actions would help successfully maintain the nurse staffing level in general wards and adequately control the COVID-19 pandemic in Korea. Appropriate plans could easily overcome any crisis of infectious diseases such as COVID-19 [39]. An empirical study conducted in the United States showed that well-planned responses of hospitals could successfully help overcome the COVID-19 pandemic [40].

Meanwhile, this study has several limitations. First, the ICUs would be the clinical department that is critically affected by the nurse staffing level in hospitals during the COVID-19 pandemic. However, this study focused on general wards. Unless reallocating or moving a large number of nurses to ICUs, identifying the direct impact of the change in nursing staff levels in general wards would be challenging. Second, this study did not use the direct measure of patients with COVID-19 by hospital; instead, a proxy variable, which is a trend of COVID-19 outbreak, was used. To detect the effect of COVID-19 on NP ratio changes, we should have used both the nurse staffing data and the direct patient data with COVID-19 by hospital. To mitigate this limitation, we analyzed the data by hospital size such as tertiary and general hospitals. Third, this study could not compare the 2020 data with the data of previous years, such as 2019. Data comparison by year would have increased the validity of our study data on whether improving the nurse staffing level in 2020 resulted from seasonal variation or any other characteristic in 2020. Unfortunately, this study did not have the opportunity to analyze the data of the previous year.

This study is important from several perspectives. First, this study would help us confirm whether hospitals had enough nurse workforces to securely supply appropriate levels of inpatient nursing care during the COVID-19 pandemic. Second, by using national-level data reporting mandatory nurse staffing levels directly from almost all the hospitals, the study obtained highly validated study results. The nursing grade data used in this study were based on reimbursement schedules and the nurse–patient (NP) ratio, which was directly reported from and objectively measured by each hospital under the national health insurance program. National data coming from an almost perfect study setting that is separated from other environments were used. No other studies used this kind of study setting. Third, this study would also provide some political grounds on whether the government made the right decision regarding the effective management of patients with COVID-19 and the nurse staffing level. This part will be discussed in the last section, using the study results. Many nations took various actions against the COVID-19 [41,42,43].

Despite these several limitations, this study has several meaningful implications for health service researchers and policymakers. First, this study is one of the first monitoring studies on the nurse staffing level and how the level changed during the COVID-19 pandemic. This study could verify and confirmed that the nurse staffing level in general wards in Korea did not decrease. Identifying the nurse staffing level of hospitals is important to maintain a certain level of quality of care. Second, this study observed the changes in the NP ratio, which was a national mandatory information that is objectively measured and reported. By using empirically collected national data, this study has high validation in terms of measuring the changes in nurse staffing levels. Third, this study used a large national dataset targeting hospital populations with more than 30 beds. The percentage of tertiary hospitals and general hospitals included in this study was 100% and 95%, respectively.

## 5. Conclusions

Through this study, we found that the nurse staffing levels of tertiary and small hospitals had improved during the COVID-19 pandemic, suggesting several political implications. First, although these changes might have resulted from seasonal variation or other unknown factors, the COVID-19 impact clearly has existed by observing various pandemic social and economic impacts. Hence, the government should start conducting research with more accurate data by linking the nurse staffing level to the number of patients with COVID-19 in hospitals. Investigating the direct effect of COVID-19 on the staffing level of ICUs is also necessary, considering that our study only investigated the changes in nurse staffing levels in the general wards of hospitals. Second, the government should also prepare for alternative action plans in dealing with nursing shortages when another infectious pandemic comes. Third, the government needs to prepare for action plans such as increasing the teleconsultation or referral systems of patients to transfer to areas with fewer COVID-19 cases to efficiently respond to the unknown impact of infectious diseases.

## Figures and Tables

**Figure 1 ijerph-18-05900-f001:**
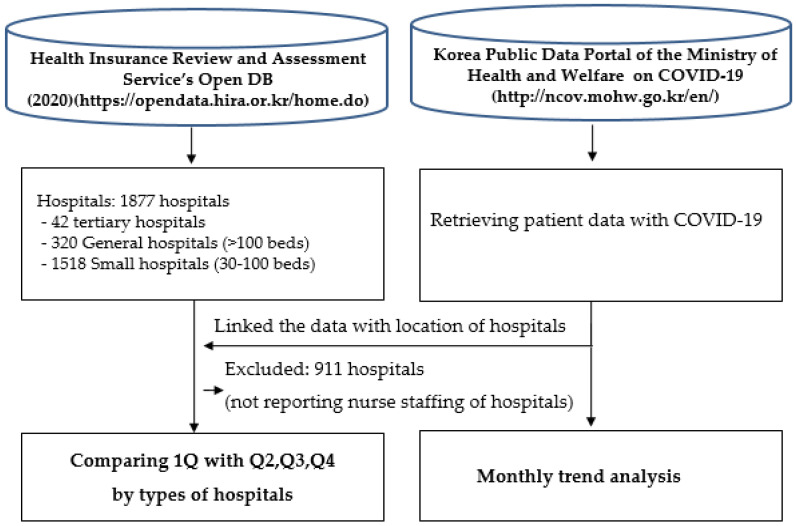
Flow of the data-processing procedure.

**Figure 2 ijerph-18-05900-f002:**
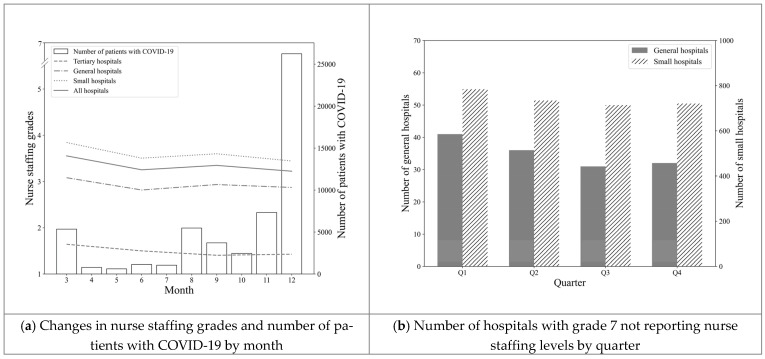
Changes in nurse staffing grades, COVID-19 cases, and number of hospitals not reporting Table.

**Table 1 ijerph-18-05900-t001:** General characteristics of the study subjects (*n* = 969).

Variables		All (*n* = 969)	Hospital Type			*p*-Value
			Tertiary (*n* = 42)	General (*n* = 268)	Small (*n* = 659)	
		*n*(%) or M ± SD	*n*(%) or M ± SD			
Operation period (years)		17.39 ± 13.29	36.64 ± 15.61	26.57 ± 13.29	12.42 ± 9.50	<0.001
Ownership	Public	110 (11.4%)	13 (31.0%)	57 (21.3%)	40 (6.1%)	<0.001
	Private	859 (88.6%)	29 (69.0%)	211 (78.7%)	619 (93.9%)	
Location	Urban	927 (95.7%)	41 (97.6%)	260 (97.0%)	626 (95.0%)	0.3193
	Rural	42 (4.3%)	1 (2.4%)	8 (3.0%)	33 (5.0%)	
Number of beds		193.5 ± 227.3	937.5 ± 386.1	323.0 ± 157.6	93.34 ± 66.44	<0.001
Number of physicians		50.76 ± 133.5	524.2 ± 330.8	78.48 ± 85.78	9.31 ± 7.65	<0.001
Herfindahl–Hirschman index		0.194 ± 0.144	0.261 ± 0.131	0.209 ± 0.150	0.184 ± 0.140	<0.001

M: mean; SD: standard deviation.

**Table 2 ijerph-18-05900-t002:** Changes in nurse staffing grades during the COVID-19 pandemic in all hospitals (*n* = 969).

Variables	Exp (β)	95% CI		*p*-Value
		LL	UL	
Hospitals: General hospitals (Ref = tertiary)	1.469	0.963	2.240	0.075
Hospitals: Small hospitals (Ref = tertiary)	1.923	1.173	3.151	0.010
Years of operation	0.999	0.994	1.004	0.714
Ownership: public(Ref = private)	0.615	0.506	0.747	<0.001
Urban location(Ref = rural)	0.857	0.638	1.151	0.304
Number of beds	0.999	0.999	1.000	<0.001
Number of physicians per 100 beds	0.988	0.983	0.993	<0.001
Herfindahl–Hirschman index	1.406	0.909	2.175	0.125
Nurse staffing grade at 2Q (Ref = 1Q)	0.728	0.624	0.849	<0.001
Nurse staffing grade at 3Q (Ref = 1Q)	0.798	0.684	0.931	0.004
Nurse staffing grade at 4Q (Ref = 1Q)	0.704	0.604	0.822	<0.001

β: regression coefficients of the generalized linear model for nurse staffing grades; Exp is the exponential function; CI: confidence interval; LL stands for lower limit and UL stands for upper limit.

**Table 3 ijerph-18-05900-t003:** Changes in nurse staffing grades during the COVID-19 pandemic in tertiary hospitals (*n* = 42).

Variables	Exp (β)	95% CI		*p*-Value
		LL	UL	
Years of operation	1.000	0.995	1.004	0.887
Ownership: public(Ref = private)	0.869	0.738	1.024	0.095
Urban location(Ref = rural)	0.683	0.417	1.119	0.132
Number of beds	1.000	0.999	1.000	0.004
Number of physicians per 100 beds	0.993	0.987	0.998	0.014
Herfindahl–Hirschman index	1.260	0.666	2.383	0.479
Nurse staffing grade at 2Q (Ref = 1Q)	0.867	0.713	1.053	0.153
Nurse staffing grade at 3Q (Ref = 1Q)	0.788	0.649	0.958	0.018
Nurse staffing grade at 4Q (Ref = 1Q)	0.807	0.664	0.981	0.033

β: regression coefficients of the generalized linear model for nurse staffing grades; Exp is the exponential function; CI: confidence interval; LL stands for lower limit and UL stands for upper limit.

**Table 4 ijerph-18-05900-t004:** Changes in nurse staffing grades during the COVID-19 pandemic in general hospitals (*n* = 268).

Variables	Exp (β)	95% CI		*p*-Value
		LL	UL	
Years of operation	0.993	0.986	1.000	0.057
Ownership: public (Ref = private)	0.598	0.477	0.751	<0.001
Urban location (Ref = rural)	0.539	0.313	0.931	0.027
Number of beds	0.997	0.997	0.998	<0.001
Number of physicians per 100 beds	0.971	0.963	0.980	<0.001
Herfindahl–Hirschman index	1.533	0.799	2.945	0.199
Nurse staffing grade at 2Q (Ref = 1Q)	0.758	0.591	0.973	0.030
Nurse staffing grade at 3Q (Ref = 1Q)	0.845	0.659	1.083	0.185
Nurse staffing grade at 4Q (Ref = 1Q)	0.792	0.618	1.016	0.067

β: regression coefficients of the generalized linear model for nurse staffing grades; Exp is the exponential function; CI: confidence interval; LL stands for lower limit and UL stands for upper limit.

**Table 5 ijerph-18-05900-t005:** Changes in nurse staffing grades during the COVID-19 pandemic in small hospitals (*n* = 659).

Variables	Exp (β)	95% CI		*p*-Value
		LL	UL	
Years of operation	0.997	0.989	1.005	0.489
Ownership: public (Ref = private)	0.304	0.216	0.430	<0.001
Urban location (Ref = rural)	0.875	0.609	1.257	0.471
Number of beds	1.005	1.004	1.006	<0.001
Number of physicians per 100 beds	1.017	1.009	1.024	<0.001
Herfindahl–Hirschman index	1.468	0.829	2.599	0.188
Nurse staffing grade at 2Q (Ref = 1Q)	0.710	0.585	0.862	0.001
Nurse staffing grade at 3Q (Ref = 1Q)	0.784	0.647	0.951	0.013
Nurse staffing grade at 4Q (Ref = 1Q)	0.671	0.553	0.814	<0.001

β: regression coefficients of the generalized linear model for nurse staffing grades; Exp is the exponential function; CI: confidence interval; LL stands for lower limit and UL stands for upper limit.

## Data Availability

The datasets used and/or analyzed in this study are available from the corresponding author on reasonable request.

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
