# Peer review of "Changes in Nurse Staffing Grades of Korean Hospitals during COVID-19 Pandemic"

_ijerph, 2021, doi:10.3390/ijerph18115900_

Round 1
Reviewer 1 Report
Thank you for the opportunity to review the paper entitled: Changes in Nurse Staffing Grades of Korean Hospitals during the COVID-19 Pandemic.
Prior to publication, the manuscript requires some improvements:
Abstract: is incomplete
Please clearly define the aim of the study: lines 16, 74, 222 are worded differently, not the same.
Introduction: No historical overview of the subject in your country, employment rate, recruitment, patient/nurse ratio and no comparison with other countries. Does not adequately justify (scarce literature) the important role of nurses.
Line 45: Wuhan
Methodology:
Does not specify the type of study.
Please explain how data were collected and data protection.
Line 89: Please clarify when approval for the project was obtained from the Bioethics Committee and what the study period was. We cannot do a study without prior approval from the Bioethics Committee.
Table 2 - not understood
Please explain the abbreviations used in all tables.
The bibliography should be revised, there are citations that do not match the journal's criteria.
I think it is a very interesting topic, not only for nurses but for all health professionals, doctors, physiotherapists, orderlies, assistants, cleaning staff, etc.... This terrible pandemic that we are suffering is going to strengthen health systems worldwide and will strengthen the union between health workers and society.
Author Response
Thanks for all of the kind words and advice.
We have revised the manuscript.

Reviewer 2 Report
Before publication the manuscript requires some improvement, details in the attached file.

Author Response

(The authors gave the same response as above.)

Round 2
Reviewer 2 Report
Accept in present form.